# Curvicollide D Isolated from the Fungus *Amesia* sp. Kills African Trypanosomes by Inhibiting Transcription

**DOI:** 10.3390/ijms23116107

**Published:** 2022-05-29

**Authors:** Matilde Ortiz-Gonzalez, Ignacio Pérez-Victoria, Inmaculada Ramirez-Macias, Nuria de Pedro, Angel Linde-Rodriguez, Víctor González-Menéndez, Victoria Sanchez-Martin, Jesús Martín, Ana Soriano-Lerma, Olga Genilloud, Virginia Perez-Carrasco, Francisca Vicente, José Maceira, Carlos A. Rodrígues-Poveda, José María Navarro-Marí, Fernando Reyes, Miguel Soriano, Jose A. Garcia-Salcedo

**Affiliations:** 1Center for Intensive Mediterranean Agrosystems and Agri-Food Biotechnology (CIAIMBITAL), University of Almeria, 04001 Almería, Spain; mog042@ual.es (M.O.-G.); msoriano@ual.es (M.S.); 2GENYO, Centre for Genomics and Oncological Research: Pfizer, University of Granada, Andalusian Regional Government, PTS Granada-Avenida de la Ilustración, 18016 Granada, Spain; inmaculada.ramirez@genyo.es (I.R.-M.); angel.linde@genyo.es (A.L.-R.); victoria.sanchez@genyo.es (V.S.-M.); ana.soriano@genyo.es (A.S.-L.); virginia.perez@genyo.es (V.P.-C.); pemaceira@gmail.com (J.M.); crodriguespoveda@gmail.com (C.A.R.-P.); 3Fundación MEDINA, Centro de Excelencia en Investigación de Medicamentos Innovadores en Andalucía, Avenida del Conocimiento 34, 18016 Granada, Spain; ignacio.perez-victoria@medinaandalucia.es (I.P.-V.); ndepedro@lifelength.com (N.d.P.); victor.gonzalez@medinaandalucia.es (V.G.-M.); jesus.martin@medinaandalucia.es (J.M.); olga.genilloud@medinaandalucia.es (O.G.); francisca.vicente@medinaandalucia.es (F.V.); 4Microbiology Unit, Biosanitary Research Institute IBS.Granada, University Hospital Virgen de las Nieves, 18014 Granada, Spain; josem.navarro.sspa@juntadeandalucia.es; 5Department of Physiology, Faculty of Pharmacy, Campus Universitario de Cartuja, Institute of Nutrition and Food Technology “José Mataix”, University of Granada, 18071 Granada, Spain

**Keywords:** African trypanosomiasis, natural products, new trypanocidal molecule, curvicollide D

## Abstract

Sleeping sickness or African trypanosomiasis is a serious health concern with an added socio-economic impact in sub-Saharan Africa due to direct infection in both humans and their domestic livestock. There is no vaccine available against African trypanosomes and its treatment relies only on chemotherapy. Although the current drugs are effective, most of them are far from the modern concept of a drug in terms of toxicity, specificity and therapeutic regime. In a search for new molecules with trypanocidal activity, a high throughput screening of 2000 microbial extracts was performed. Fractionation of one of these extracts, belonging to a culture of the fungus *Amesia* sp., yielded a new member of the curvicollide family that has been designated as curvicollide D. The new compound showed an inhibitory concentration 50 (IC_50_) 16-fold lower in *Trypanosoma brucei* than in human cells. Moreover, it induced cell cycle arrest and disruption of the nucleolar structure. Finally, we showed that curvicollide D binds to DNA and inhibits transcription in African trypanosomes, resulting in cell death. These results constitute the first report on the activity and mode of action of a member of the curvicollide family in *T. brucei*.

## 1. Introduction

African trypanosomiasis is caused by the flagellated protozoa parasite *Trypanosoma brucei*. Two subspecies, *T. b. gambiense* in central and western Africa and *T. b. rhodesiense* in eastern and southern Africa, infect humans causing human African trypanosomiasis (HAT). The third subspecies, *T. b. brucei*, causes animal trypanosomiasis or Nagana, in the livestock [1]. The parasite is transmitted through the bite of an infected tsetse fly during a blood meal. Once in the skin of the mammalian host, the parasites rapidly multiply in the blood and eventually colonize the brain. There are two stages of the disease, the early haemo-lymphatic stage and the late meningo-encephalitic stage, when parasites invade the central nervous system [1]. Sleeping sickness threatens millions of people in 36 countries in sub-Saharan Africa. According to the World Health Organization (WHO), from 2016 to 2020, 55 million people were at risk of suffering HAT by *T. b. gambiense* and about 3 million by *T. b. rhodesiense* [2].

Nowadays, new reported cases tend to decrease over the years, but because African trypanosomiasis mainly affects remote rural communities in regions with poor health infrastructures, many cases are undiagnosed or unreported, and the real burden of the disease in Africa remains unknown [1]. Additionally, trypanosomiasis also affects domestic animals, especially cattle, being a major obstacle to the economic development of the affected rural areas. The sub-Saharan region records an annual loss that ranges from 1.5 to 5 billion U.S. dollars, deriving from livestock being killed by African trypanosomes [3].

No vaccines are available to date due to the ability of the parasite to evade the host immune system by changing the variant surface glycoprotein (VSG); thus, the treatment of the disease relies exclusively on chemotherapy. Few drugs are currently available and most of them were developed more than half a century ago. The election of the drug is based on the disease stage and causative pathogen. For the first stage of the disease, either pentamidine or suramin is used for HAT caused by *T. b. gambiense* and *T. b. rhodesiense,* respectively [4]. For the second stage of the disease, the organoarsenic compound melarsoprol is currently the only effective drug against both subspecies. In 2009, nifurtimox, a drug used to treat American trypanosomiasis (Chagas disease), was introduced as a combination therapy with eflornithine, to treat second stage infections caused by *T. b. gambiense*. All of them show limitations, from high toxicity and drug resistance to poor efficacy [5]. Efforts have been made to overcome these drug-associated resistances [5,6], but there is an urgent and global need for new drugs with new modes of action.

Recently, a new drug, fexinidazole, has been added to the list of treatments. Fexinidazole is the first oral treatment for stage 1 and stage 2 HAT produced by *T. b. gambiense*. It was approved for use by the European Medicines Agency (EMA) in December 2018 [7]. However, it has been recently reported that it is less effective (91%) than nifurtimox–eflornithine combination therapy (98%) in the treatment of patients in the late-stage of the disease [8,9]. In addition, in patients showing more than 1000 white blood cells per milliliter in the cerebrospinal fluid, the risk of failure was greater with fexinidazole and the EMA recommendation for these patients is the use of fexinidazole only if no other suitable treatment is available or tolerated [10]. Overall, administration of fexinidazole has significantly improved the therapeutic options for the disease. However, there is a need to find additional treatments that overcome the limitations of fexinidazole.

The biggest challenge for the discovery of new drugs is the identification of molecules with pharmacological activity and tolerable toxic profile. Natural products are structures that have been synthesized, degraded and transformed by enzymatic systems and that have traditionally been used as the main source of new drugs [11,12,13]. 

In this study, we performed a high-throughput screening (HTS) of a natural products library derived from microbial extracts from fungi and Actinobacteria, to identify new molecules with trypanocidal activity.

## 2. Results

### 2.1. Isolation and Structural Characterization of Curvicollide D

As a results of a high-throughput screening (HTS) of a collection of 2000 microbial extracts, we selected one fungal extract with high trypanocidal activity, belonging to a culture of the fungus *Amesia* sp. This extract yielded a new member of the curvicollide family that we have designated as curvicollide D (Figure 1).

Curvicollide D was isolated from a bioactive acetone extract of the strain CF-258252 cultured in BRFT medium and purified through SP207ss column chromatography and semipreparative reversed-phase C8 high performance liquid chromatography (HPLC). A molecular formula of C_26_H_40_O_5_ was assigned to the compound based on the observed ion [M + NH_4_]^+^ at m/z 450.3217 (calcd. for C_26_H_44_NO_5_^+^, 450.3214) in its (+)-ESI-TOF spectrum (Appendix A) and its isotopic pattern. The obtained molecular formula was entered as query in the Dictionary of Natural Products (CRC Press “Dictionary of Natural Products on USB, ver.30.2”) rendering 36 candidates which were reduced to just five if considering only metabolites of fungal origin. Two of those compounds, curvicollides A and B, display a single absorption maximum at 253 nm (neutral conditions in CH_2_Cl_2_), very close to the single maximum at 244 nm observed in the UV (DAD) spectrum of the target compound retrieved from the LC-DAD-HRMS analysis (Appendix A). The ^1^H NMR spectrum of molecule (Table 1, Appendix A) was remarkably similar to that of curvicollides A and B, with just minor differences that identified it as a new member of the curvicollide family. Additional 1D ^13^C (Table 1, Appendix A) and 2D NMR spectra (including Correlated Spectroscopy (COSY), Total Correlation Spectroscopy (TOCSY), Nuclear Overhauser Effect Spectroscopy (NOESY), (Heteronuclear Single Quantum Correlation) (HSQC) and Heteronuclear Multiple Bond Correlation (HMBC)) (Appendix A) were recorded to fully elucidate its structure. Interpretation of the ^1^H, HSQC and HMBC spectra revealed the presence of 5 quaternary carbons (including one ketone carbonyl at *δ*_C_ 213.2, one ester carbonyl at *δ*_C_ 176.2, two olefinic carbons in the range *δ*_C_ 130–139, and one oxygenated carbon at *δ*_C_ 73.1), 11 methines (including six olefinic carbons in the range *δ*_C_ 120–144, two oxygenated methines at *δ*_C_ 75.1 and *δ*_C_ 90.4 and three aliphatic methines in the range *δ*_C_ 42–53), 3 aliphatic methylenes in the range *δ*_C_ 23–43 and finally 7 methyls (one triplet corresponding to an aliphatic chain end, two doublets and four singlets). Detailed analysis of the COSY and TOCSY correlations identified different spin systems which could be connected using the key long-range correlations observed in the HMBC spectrum rendering the final connectivity of the molecule (Figure 2). The determined carbon skeleton confirmed the anticipated curvicollide nature of the compound, corroborated through further comparisons with the NMR data of curvicollides A and B. Curvicollides display a hydroxy group in the C15–C20 segment of the compounds, at C16′ in curvicollide A (rendering a primary alcohol functional group) and at C19 in curvicollide B (rendering a secondary alcohol functional group). Curvicollide D has such hydroxy group as a C16 substituent, rendering a tertiary alcohol functional group. The relative stereochemistry at the C3–C4 and the C9–C11 stereoclusters was determined based on the observed key NOEs and coupling constants (Figure 3). In the non-polar CDCl_3_ solvent, the intramolecular hydrogen bond between the hydrogen atom of the hydroxyl group at C4 and the oxygen carbonyl at C2 favors a main chair-like conformation for the C1–C5 segment that enables the application of the Stilles-House method [14,15] which renders a *threo* relative configuration for the chiral centres at positions 3 and 4 (Figure 3). This *threo* configuration was also in agreement with the ^13^C chemical shifts for the C4 carbinol, the C3 methine and the C3′ methyl, according to the empirical rule of Heathcock [16]. The relative stereochemistry of the lactone ring was established based on the key NOESY correlations and the coupling constants *J*_H9H10_ and *J*_H10H11_, as already described for curvicollides A–C [17] (Figure 3). The determined relative configuration of each of these stereoclusters turned out to match that reported for curvicollide C [18]. The essentially identical NMR data for the C1–C14 segment of curvicollides A–C and the new compound confirms that the mentioned stereoclusters display also the same stereochemical relationship in the new compound and curvicollides A–C. It can be safely assumed that the absolute stereochemistry of the chiral centers at positions 3, 4, 9, 10 and 11 is also identical since the new molecule and the previously reported curvicollides must be the products of homologous biosynthetic gene clusters. The absolute configuration of the remaining chiral centre at position 16 could not be determined since the tertiary alcohol at that position is too hindered to be derivatized with chiral Mosher’s reagent and the isolated amount was insufficient for attempting crystallization. The new compound has been designated as curvicollide D (Figure 1).

### 2.2. Curvicollide D Inhibits the Growth of T. brucei Bloodstream Forms and Causes Cell Cycle Arrest at G_2_/M Phase

The trypanocidal effect of the purified curvicollide D was evaluated by resazurin assay. Curvicollide D showed a dose-dependent effect on *T. brucei* bloodstream forms growth, with half-maximal inhibitory concentration (IC_50_) value of 1 + 09 μM after 24 h of incubation (Figure 4A). At this concentration, no effect was observed in human hepatocarcinoma (Hep G2) cells. The IC_50_ of curvicollide D in Hep G2 cells was 16 + 01 μM (Figure 4B), 16-fold higher than in *T. brucei* bloodstream forms, yielding a selectivity index of 16. The IC_75_ of the compound against *T. brucei* (1.5 μM) was chosen to carry out the rest of this study.

To analyze the effect of curvicollide D on cell cycle progression, we assessed the DNA content by flow cytometry. In untreated trypanosomes, the distribution of cells in each cell cycle phase, as the percentage in relation to the total number of cells, showed the typical distribution for bloodstream trypanosomes, with 50% of cells at G1 phase, 12% at S phase and 38% at G_2_/M [19] (Figure 4C,D). However, trypanosomes treated with curvicollide D showed an increased percentage of cells in the S and G_2_/M phases after 6 h of treatment (15% and 42% respectively). These changes were even more remarkable after 15 h, with a significant increase in the G_2_/M population (64%) and a significant decrease in the % of cells in at G1 phase (10%) (Figure 4C,D). 

### 2.3. Curvicollide D Induces Morphological Changes and Nucleolus Disassembly in T. brucei

We next investigated the morphological changes produced by curvicollide D treatment in *T. brucei* by confocal microscopy using the DNA specific dye DAPI. Within 6 h, about 50% of the cells exhibited an abnormal morphology and after 16 h of incubation, most of the cells no longer possessed the typical long slender morphology (Figure 5A,B). The effect of curvicollide D on trypanosome morphology resulted in an enlargement/swelling of the cell posterior end that progressed along time (Figure 5B). After 3 h of treatment, no significant differences in morphology were noted, although a striking effect was detected at the DNA level (Figure 5C). Within the trypanosome cell, the nucleus can be visualized as a large stained spot and the kinetoplast (mitochondrial DNA) as a smaller focus. The nucleolus is clearly visible as an unstained area in the center of the nucleus. After 3 h of incubation with 1.5 μM curvicollide D this unstained region in the nucleus was no longer appreciable in 90% of the cells (Figure 5C,D), which suggested the disassembly of the nucleolus. Nucleolus disruption was next confirmed by immunofluorescence with L1C6 anti-nucleolar antibody (Figure 5E). After treatment, the green signal of the antibody decrease significantly and/or spread through the nucleus [20].

### 2.4. Curvicollide D Inhibits T. brucei RNA Polymerases I and II

The main function of the nucleolus in eukaryotic cells is the transcription of the ribosomal genes, a process accomplished by RNA polymerase I (Pol I) that must be tightly regulated to achieve proper cellular proliferation and cell growth [21]. Therefore, nucleolus disassembly is a cellular hallmark of rRNA transcription blockage.

To determine whether curvicollide D affected Pol I transcription, we quantified the short-lived external transcribed spacer (ETS) region of the pre-rRNA, whose abundance is reflective of the rRNA synthesis rate. For this purpose, we performed real time quantitative reverse transcription PCR (qRT-PCR) analysis using two sets of primers covering two distant regions within the rRNA locus (Figure 6A). Incubation of trypanosomes with 1.5 μM curvicollide D for 3 h significantly repressed the expression of the rRNA precursor transcripts, as detected with actinomycin D, a known inhibitor of transcription (Figure 6B). No significant differences were found between the two distant (1) and (2) primer sets.

In African trypanosomes, Pol I also transcribes the active expression site (ES) (Figure 6C): a subtelomeric locus that contains the codifying gene for the major variable surface glycoprotein (*VSG*), which covers the surface of the bloodstream-form [22]. Although there are around 20 ESs, only one is expressed at each time [23], which is crucial for the evasion of the host immune response by antigenic variation. All of them have a similar structure; they are 60 kb long polycistronic transcription units, encoding for different expression site associated genes (*ESAGs*), followed by a 70 bp long region of repetitive DNA and the *VSG*, which is always found immediately adjacent to a telomere [22]. In the *T. brucei* strain used in this study, the studied active expression site is the one that transcribes the 221*VSG* gene (Figure 6C). Like in the ribosomal locus, we studied the effect of curvicollide D on the transcription of the active expression site 221 using two pairs of primers: one near the promoter and a second one after *ESAG6* gene (Figure 6C). Treatment with curvicollide D for 3 h resulted in a strong reduction of the *VSG221* ES precursor as estimated by qRT-PCR (Figure 6D). No significant differences were found in the inhibition rate between the primer sets (3) and (4).

Next, we investigated whether curvicollide D also affected Pol II-mediated transcription. In African trypanosomes, Pol II transcribes mRNAs as well as the spliced-leader RNA genes (SL-RNA). SL-RNA is a monocistronic noncoding Pol II-transcribed RNA that is trans-spliced to the 5′-end of all trypanosome mRNAs, conferring the 5′-cap structure required for RNA maturation, export and translation [24]. These are the only genes in this parasite with a defined Pol II promoter [25,26]. To analyze the effect of curvicollide D on Pol II transcription in *T. brucei* we quantified the abundance of *SL* precursor transcripts using two pairs of primers located in intergenic regions of the SL-RNA locus, downstream of the Pol II promoter (Figure 6E). α-amanitin was used as a positive control for the inhibition of Pol II transcription. A strong reduction in the levels of SL-RNA precursor transcripts was detected after treatment with curvicollide D for 3 h. The inhibition was greater than that caused by α-amanitin and similar to Pol I inhibition (Figure 6F). In summary, these results demonstrate that curvicollide D inhibits Pol I and Pol II transcription in African trypanosomes.

### 2.5. Curvicollide D Binds to Duplex DNA and Displaces Ethidium Bromide

Several compounds have been described as transcription inhibitors because of their ability to intercalate into double-strand DNA [27,28]. Fluorescent intercalator displacement (FID) assay has been extensively used to identify DNA binding compounds [29,30]. This assay relies on the displacement of a fluorescent DNA-binding ligand by the compound of interest. In the presence of DNA and an intercalating fluorophore, the introduction of a competing putative DNA binding agent results in a differential decrease in fluorescence. To test whether curvicollide D binds to duplex DNA, we performed a FID assay using ethidium bromide (EtBr) as a fluorescent intercalator and plasmid DNA as a template. A concentration of 5 μM EtBr yielded a fluorescence peak at 610 nm (black line Figure 7). Upon addition of increasing concentrations of curvicollide D (from 10 to 30 μM), we observed a progressive decrease in the fluorescence intensity (from 10% to 50%) (Figure 7A,B). These results indicated that curvicollide D competes with ethidium bromide binding sites in duplex DNA.

## 3. Discussion

Most of the existing treatments for HAT in the 21st century are over 40 years old. Although they have saved hundreds of thousands of lives, the majority of them have lost their initial efficacy and have generated resistance in certain regions, where they are currently not effective. In this study, our aim was to identify non-previously reported molecules from natural products with trypanocidal activity. 

Nature has evolved to produce a wide variety of natural products with unique biological activity in terms of target affinity and specificity. Among them, antibiotics are the most popular example [31]. Since the identification in 1940 of the antibacterial properties of penicillin by Fleming [32], the screening of microorganisms, particularly soil actinobacteria and fungi, has progressively increased in an attempt to identify other molecules for the treatment of diseases [33]. The structural diversity of these small molecules has allowed the identification of new leading compounds produced by diverse soil taxa by HTS [34]. Currently, around 60% of drugs available on the market are natural products or their structural analogues [27,34,35,36].

Based on these premises, we have performed a HTS on a collection of natural products belonging to Fundación MEDINA (Granada, Spain) [37,38]. As a result of this process, we have isolated a new member of the curvicollide family from a culture of *Amesia* sp. strain CF-258252, curvicollide D. The structure of this new natural product has been elucidated after extensive analysis of its HRMS and 1D and 2D NMR spectra and it corresponds to an isomer of curvicollides A and B, where one of the two hydroxy groups in the molecule is located at C-16 in curvicollide D, whereas it is located at C-16′ or C-20 in curvicollides A and B, respectively.

Curvicollides A-C were initially discovered from a *Podospora curvicola* isolate. All of them displayed a significant antifungal activity against *Aspergillus flavus* and *Fusarium verticillioides* [17], but the mode of action behind this activity remains unknown to date. 

In this study, we have shown that curvicollide D has a potent trypanocidal activity against *T. brucei* bloodstream forms, with half-maximal inhibitory concentration (IC_50_) value of 1 μM, 16-fold lower than in Hep G2 human cells. Thus, its selective index, >16, meets the DNDi standards for the selection of drugs for the treatment of diseases caused by kinetoplastid parasites. The criterium specifies that IC_50_ values should be ≤10 μM and had a selectivity index of ≥10 [39].

Treatment of *T. brucei* with curvicollide D led to cell cycle arrest at G_2_/M phase followed by cell death. The compound also induced changes in the DAPI staining pattern of the nucleus, suggesting the disruption of the nucleolus structure. This observation was later verified by the delocalization of the nucleolar marker protein L1C6 [20] after 3 h of treatment.

The nucleolus disassembly suggested that curvicollide D could be affecting the nucleolar function. This hypothesis was confirmed by analysis of ribosomal DNA transcription performed by Pol I. Transcription of the short-lived external transcribed spacer (ETS) region in the pre-rRNA was quantified using qRT-PCR, showing reduction of 90%. Moreover, transcription of the active VSG expression site, which is also transcribed in *T. brucei* by Pol I was reduced by 80%. In addition, a significant decrease in the precursor transcript of the splice leader gene, transcribed by RNA Pol II, was observed after treatment. These results indicate that curvicollide D is a general transcription inhibitor. Finally, using FID assay, we demonstrated that curvicollide D intercalates into DNA displacing the complex formed by EtBr-DNA. 

Clinically, DNA binding agents have been used as antiparasitic, antibacterial or antitumor agents [40]. Mepacrine or acridine derivatives (mepacrine, proflavine and ethidium bromide) have been used in the treatment of trypanosomiasis [41]. Specifically, EtBr was used to treat cattle African trypanosomiasis [42], and diminazene aceturate, another DNA binding agent, was used for the same purpose [43]. In this line, other natural products, such as camptothecin, etoposide or doxorubicin, which intercalate into the DNA, also exert a cytotoxic effect through cell cycle arrest [44,45,46,47].

In African trypanosomes, selective inhibition of transcription has been shown to be a suitable target due to their high proliferation rate [48]. A previous study, demonstrated that compounds that bind to DNA and inhibit RNA Pol I transcription, such as BMH-21 [49], CX-3543 [50], and CX-5461 [51], showed trypanocidal effect at low concentrations, inducing cell cycle arrest at G2/M phase and affecting nucleolus structure as we observed with curvicollide D [52].

Herein, we described for the first time the mode of action of a new member of the curvicollide family, curvicollide D, which consists of an inhibition of transcription through DNA intercalation. In addition, this new trypanocidal compound could constitute the starting point to develop new leading compounds for new therapies against *T. brucei*. It is important to emphasize that the synthesis of the structurally similar curvicollide C has already been achieved [18]. This accomplishment makes feasible the commercial synthesis of other members of the curvicollide family and their derivatives with potential trypanocidal activity.

## 4. Materials and Methods

### 4.1. General Experimental Procedures

Solvents employed were all HPLC grade. LRMS analyses were performed on an Agilent 1260 Infinity II (Agilent Technologies, Santa Clara, CA, USA) single quadrupole LC-MS system. HRESIMS and MS/MS spectra were acquired using a Bruker maXis QTOF mass spectrometer (Bruker Daltonik GmbH, Bremen, Germany) coupled to an Agilent 1200 Rapid Resolution HPLC. Medium pressure liquid chromatography (MPLC) was performed on a CombiFlash Teledyne ISCO Rf400x apparatus (Teledyne ISCO, Lincoln, NE, USA). Preparative or semi-preparative HPLC purifications were performed on a Gilson GX-281 322H2 HPLC (Gilson Technologies, Middleton, WI, USA). NMR spectra were recorded at 297 K on a Bruker Avance III spectrometer (500 and 125 MHz for ^1^H and ^13^C, respectively) equipped with a 1.7 mm TCI MicroCryoProbe^TM^ (Bruker Biospin, Fällanden, Switzerland). ^1^H and ^13^C chemical shifts were reported in ppm using the signals of the residual solvents as internal reference (*δ*_H_ 7.26 and *δ*_C_ 77.0 ppm for CDCl_3_). 

### 4.2. Isolation and Characterization of the Producing Strain

The producing microorganism CF-258252 was isolated from a soil sample collected in Sabino Canyon, Coronado National Forest, Arizona (USA) by the ethanol pasteurization method using 90 mm petri dishes with PCA medium [53]. Frozen stock cultures in 10% glycerol (−80 °C) are maintained in the fungal collection of Fundación MEDINA. Furthermore, to estimate the approximate phylogenetic position of strain CF-258252, genomic DNA was extracted from mycelia grown on malt-yeast extract agar. DNA extraction, PCR amplification and DNA sequencing were performed as previously described by Gonzalez-Menendez [54]. Sequences of the complete ITS1-5.8S-ITS2 and initial 28S region or independent ITS and partial 28S rDNA sequences were compared with sequences deposited at GenBank^®^ or the NITE Biological Resource Center [55] by using BLAST^®^ application [56]. Database matches yielded a very high sequence similarity (99%) to the strains *Amesia gelasinospora* CBS 673.80T, CBS 643.83 and NBRC32880, thus indicating that strain CF-258252 showed affinities to genus *Amesia* (Chaetomiaceae). Similarly high scores to other authentic fungi strains, e.g., *A. atrobrunea* CBS 379.66T (98%) *A. nigricolor* CBS 600.66T (97%), indicated that CF-258252 can be classified as *Amesia* sp.

### 4.3. Fermentation Process

Fermentation by *Amesia* sp. CF-258252 was performed through the inoculation of ten mycelial agar plugs into a flask containing SMYA medium (Bacto neopeptone 10 g; maltose 40 g; yeast extract 10 g; agar 3 g; H_2_O 1 L), 50 mL medium in a 250 mL Erlenmeyer. The flask was incubated on a rotary shaker at 220 rpm and 22 °C with 80% relative humidity. After growing the seed stage for 7 days, a 4.5 mL aliquot was used to inoculate each flask of the production medium BRFT [57]. The 15 flasks (100 mL medium in 500 mL nonbaffled flask) were incubated and 22 °C with 80% relative humidity in static condition for 21 days.

### 4.4. Isolation of Curvicollide D

The fermentation broth was thoroughly mixed with an equal volume of acetone and shaken for 2 h in a Kühner shaker. The biomass was decanted and the acetone content in the supernatant was reduced under a nitrogen stream until the volume was ca. 1 L. This crude acetone extract was charged (while mixing with water in an equal ratio) on a SP207ss resin-containing column (dimensions). After washing the loaded column with 1 L of water, the elution was performed in a CombiFlash using a linear acetone gradient from 0–100% and collecting fractions of 15 mL. Fractions active in the *Trypanosome* bioactivity assay were pooled and further fractionated by semipreparative reversed-phase HPLC (Zorbax SB-C8, 9.4 × 250 mm, 5 μm, 3.6 mL/min) using H_2_O (solvent A) and CH_3_CN (solvent B). Elution was carried out using isocratic conditions of 5% solvent B for 1.5 min and then a linear gradient from 5% to 100% solvent B in 30.5 min to afford curvicollide D (1.5 mg, *t*_R_ = 25 min), responsible for the observed antiparasitic activity.

Curvicollide D: white amorphous solid; ^1^H and ^13^C NMR data see Table 1, (+)-ESI-TOF *m*/*z* 450.3217 [M + NH_4_]^+^ (calcd for C_26_H_44_NO_5_^+^, 450.3214, Δ = 0.7 ppm), 415.2844 [M + H-H_2_O]^+^ (calcd for C_26_H_39_O_4_^+^ 415.2843, Δ = 0.2 ppm), 397.2739 [M + H-2H_2_O]^+^ (calcd for C_26_H_37_O_3_^+^ 397.2737, Δ = 0.5 ppm), 882.6105 [2M + NH_4_]^+^ (calcd for C_52_H_44_NO_10_^+^, 882.6090, Δ = 1.7 ppm), and 887.5660 [2M + Na]^+^ (calcd for C_52_H_44_NO_10_^+^, 887.5644, Δ = 1.8 ppm).

### 4.5. Cell Culture

*T. b. brucei*, Lister 427 bloodstream-form MITat 1.2 (clone 221), were cultured in HMI9 medium (GIBCO) supplemented with 10% of inactivated fetal bovine serum (iFBS) (Capricorn, FBS-11A), at 37 °C and 5% CO_2_ atmosphere. Cells derived from human hepatocarcinoma (Hep G2) were cultured in Eagle’s Modified Essential Medium (MEM) supplemented with 10% iFBS, 2 mM L-glutamine, 1 mM sodium pyruvate, and 100 µM MEM nonessential amino acids (ThermoFisher Scientific, Waltham, MA, USA, 11140) and incubated at the same conditions as parasites. 

### 4.6. Resazurin-Based Assay

Trypanocidal assays started with an initial cellular density of 5 × 10^2^ and 1 × 10^3^ cells/well of bloodstream forms in 384 and 96 well plates, respectively. In the case of 384 well plates: 45 µL of parasite culture was added to 384-well corning assay plates already containing 5 µL of the microbial extracts (final concentration of 1/300) and controls and incubated for 20 h at 37 °C. Then 10 µL of Resazurin (Sigma-Aldrich/Merck KGaA, Dramstadt, Germany, R7017) at working concentration of 0.5 mM was added per well and plates were further incubated for 4 h at 37 °C. The final fluorescence was determined at 550–590 nm. Pentamidine isethionate salt (Sigma-Aldrich, P0547) and drug free HMI9 medium, were used as negative and positive growth controls, respectively. For 96-well plates we followed the same protocol, but the final volume of the reagents and numbers of parasites were doubled. The half-maximal inhibitory concentration (IC_50_) value was determined in 96-well plates, using ten serial dilutions from an initial concentration of 50 μM as previously described [58]. Cytotoxicity in the hepatocarcinoma cell line Hep G2 was assessed using the same protocol and 10^5^ cells per well.

### 4.7. Analysis of Cell Cycle

The DNA content was assessed by fluorescence-activated cell sorting (FACS) as previously described [19]. Control samples or after treatment with 1.5 μM of curvicollide D were fixed in 900 µL of ice cold ethanol 70% and incubated in ice for 5 min. Then, cells were washed with phosphate-buffered saline (PBS) pH 7 and incubated with propidium iodide (PI) (Sigma-Aldrich, P4864) staining solution (PBS containing PI 40 µg/mL and RNAase A 100 µg/mL (Quiagen, Hilden, Germany, 19101) at 37 °C for 30 min. The analysis was performed in a FACSCanto II flow cytometer. FlowJo softward was used to estimate the number of cells at G1, S and G_2_/M phases of the cell cycle. The experiment was performed in triplicate.

### 4.8. Confocal Microscopy

Bloodstream forms of *T. brucei* treated with curvicollide D or dimethyl sulfoxide (DMSO) (control) were washed with PBS at pH 7.5. Parasites were fixed in 4% (*w*/*v*) paraformaldehyde (PFA) (Sigma-Aldrich, P6148) and permeabilized with a solution containing 0.2% triton X-100 (Sigma-Aldrich, T8787) in PBS buffer as previously describe [59]. For immunofluorescence, fixed cells were treated with the primary antibody anti L1C6 (mouse) (kindly provided by Keith Gull) in blocking buffer containing 3% bovine serum albumin (Sigma-Aldrich, A7906) during 1 h at room temperature. Samples were then incubated with the secondary antibody goat anti-mouse IgG Alexa 488 (Invitrogen, Waltham, MA, USA, A11001) in the same blocking buffer. On the other hand, a group of fixed cells treated with curvicollide D or DMSO were only coated in a poly-L-lysine-coated slides (Thermo Fisher Scientific, Waltham, MA, USA, J2800AMNZ) and mounted in DAPI-containing Vectashield (H-1200, Vector) (Vector laboratories, Burlingame, CA, USA). Image acquisition was performed with a confocal LSH 710 Confocal Microscope (Zeiss) and image analysis with ZEN blue (Zeiss) software.

### 4.9. RNA Isolation and Reverse Transcription

Total RNA was isolated from bloodstream forms of *T. b. brucei* following the TRIzol protocol (Invitrogen, 15596). Reverse transcription was performed using RevertAid First Strand cDNA Synthesis Kit (Thermo Fisher Scientific, K1621) with random primers. The reactions were incubated initially at 25 °C (5 min) and then at 42 °C for one hour. To stop the reaction, the mixture was heated at 70 °C for 5 min.

### 4.10. q-PCR Experiments

RNA transcript analysis was performed using quantitative reverse transcription PCR (q-PCR) on the 7500 Fast Real-Time PCR System (Life Technologies, Carlsbad, CA, USA) with SYBR Green (Thermo Fisher Scientific, 4309155). The reaction mix volume (10 μL) included 0.5 μM of each primer and 1 μL of cDNA diluted at 200 ng/μL. Amplification conditions were: 40 cycles of 50 °C, 2 min; 95 °C, 10 min; 95 °C, 15 seg; 60 °C, 1 °C min), and the primer sequences are listed below. Transcript levels were normalized against the number of parasites and plotted as relative change to the untreated cells. Analysis of all samples were performed in triplicate. Actinomicin D (Sigma-Aldrich, A9415) and α-amanitine (Sigma-Aldrich, A2263) were used as positive controls of inhibition for Pol I and Pol II transcription respectively. 

#### Primers Used for q-PCR

Primers used in qPCR experiments in Figure 6, were purchased from Sigma-Aldrich. Sets of primers (1) and (2) are located in the ribosomal DNA. Primers (3) and (4) are located within the active expression site VSG221. Sets of primers (3) and (4) are located within the splice leader locus. All primers were located in intergenic regions, (see Table 2).

### 4.11. Fluorescence Intercalator Displacement Assay (FID)

Ethidium bromide (EtBr) (Sigma-Aldrich, E1510) displacement assay was performed in 96-well, flat bottom polystyrene microplates (Thermo Fisher Scientific, 130188) as previously described [30]. Each well was loaded with Milli-Q water, 5 μM of ethidium bromide, 1 μM of duplex DNA (pMES4 plasmid) (GenBank: GQ907248.1) and increasing concentrations of curvicollide D (10 μM, 20 μM, 25 μM and 30 μM). The excitation wavelength of ethidium bromide was set at 510 nm and the emission profile was monitored from 520 to 720 nm with a multi-plate reader. Experiments were conducted in triplicate at room temperature (20 °C). % fluorescence values were calculated as follows: % Fluorescence = A/B × 100, where A is the fluorescence value in presence of curvicollide D and B is the fluorescence value in curvicollide D-free controls.

## Figures and Tables

**Figure 1 ijms-23-06107-f001:**
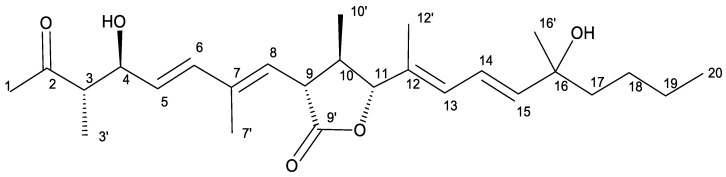
Structure of curvicollide D.

**Figure 2 ijms-23-06107-f002:**
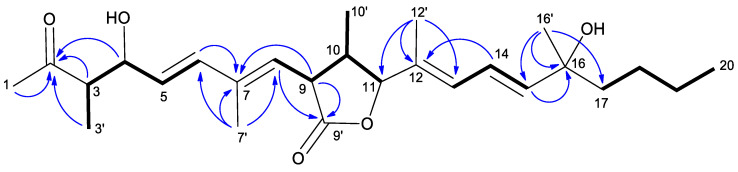
Key COSY and TOCSY correlations (bold bonds) determining the different spin systems of curvicollide D. Key HMBC correlations (blue arrows, H to C) connecting the different spin systems and rendering the connectivity of curvicollide D.

**Figure 3 ijms-23-06107-f003:**
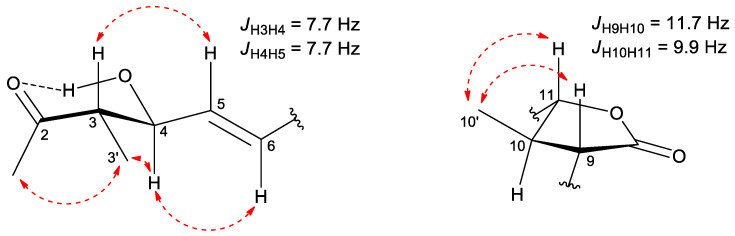
Key NOESY correlations (red dashed arrows) and coupling constants determining the relative stereochemistry of the stereoclusters C3–C4 and C9–C11 in curvicollide D.

**Figure 4 ijms-23-06107-f004:**
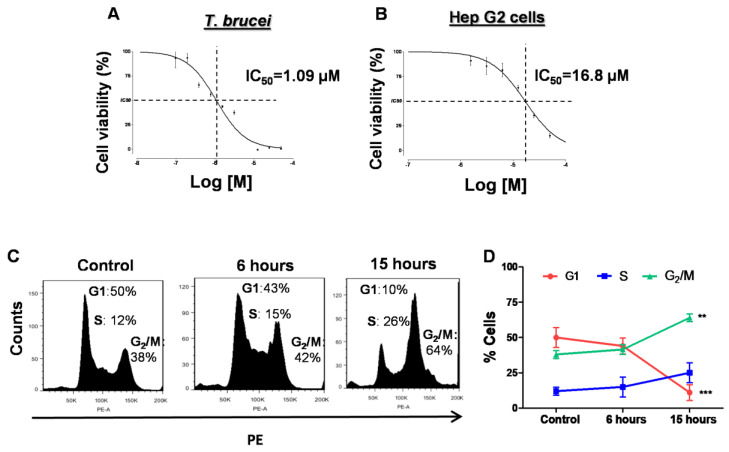
Curvicollide D inhibits proliferation and affects the cell cycle of bloodstream form of *T. brucei*. Resazurin in vitro cytotoxicity assay was used to determine the curve dose response (viability percentage versus curvicollide D concentration). The IC_50_ was determined interpolation, using GraphPad software (**A**) IC_50_ of *T. brucei* and (**B**) IC_50_ of the human hepatocarcinoma cells (Hep G2) during 24 h. (**C**,**D**) Flow cytometry analysis of the cell cycle after 6 and 15 h of exposure to curvicollide D (1.5 μM) or after treatment with DMSO (control). Percentages for each phase were determined using FlowJo software. (PE) phycoerythrin. Data represent mean and standard deviation from three independent experiments. Statistical significance between control and treatmens at 6 and 15 h was estimated using Student’s two-tailed *t*-test. *p*-Values were expressed as follows: ** *p* < 0.01 and *** *p* < 0.001.

**Figure 5 ijms-23-06107-f005:**
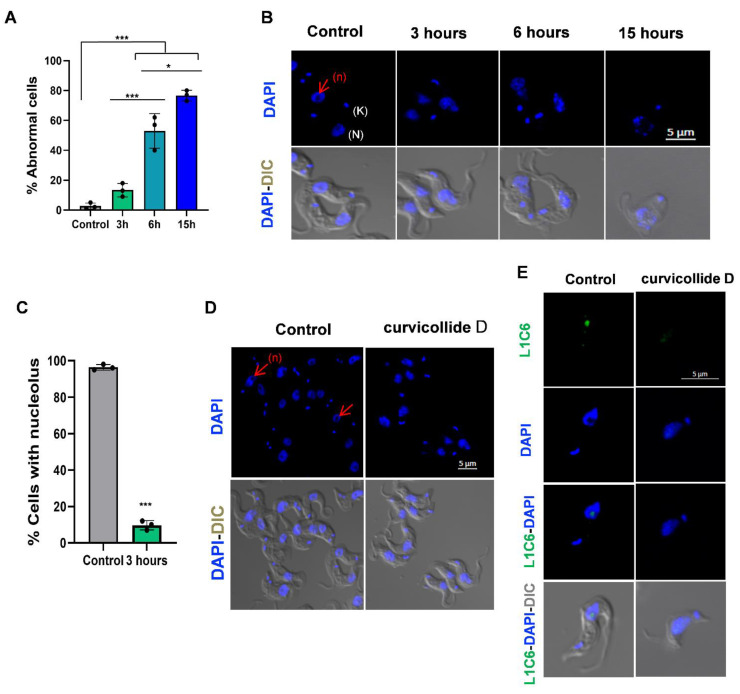
Changes in cell morphology and nucleolar structure after 3, 6 and 15 h of treatment with curvicollide D. Bloodstream forms of *T. brucei* incubated with 1.5 μM of Curvicollide D for 3, 6 and 15 h and cells treated with DMSO (control). (**A**) Percentage of abnormal cells. Error bars represent the standard deviations from three independent experiments. Statistical significance was *, *p* < 0.05; ***, *p* < 0.001 using ANOVA and Bonferroni post hoc test. (**B**) Changes in cell morphology after 3, 6 and 15 h of treatment. DAPI-stained parasites were observed by confocal microscopy (top line) and so was differential interference contrast (DIC) with DAPI (bottom line). The nucleus (N) and kinetoplast (K) are shown in white. The arrow in red indicates the nucleolus (n). (**C**) Percentage of cells that maintain the nucleolus structure after treatment. (**D**) Confocal microscopy showing disassembly of the nucleolus after a 3 h treatment with curvicollide D. (**E**) Immunofluorescence microscopy using an antibody against the nucleolar marker L1C6 (green) after a 3 h treatment with curvicollide D. Scale bar, 5 μm. Statistical significance between control and 3 h of treatment was estimated using Student’s two-tailed *t*-test. *p*-Value was expressed as follows: *** *p* < 0.001.

**Figure 6 ijms-23-06107-f006:**
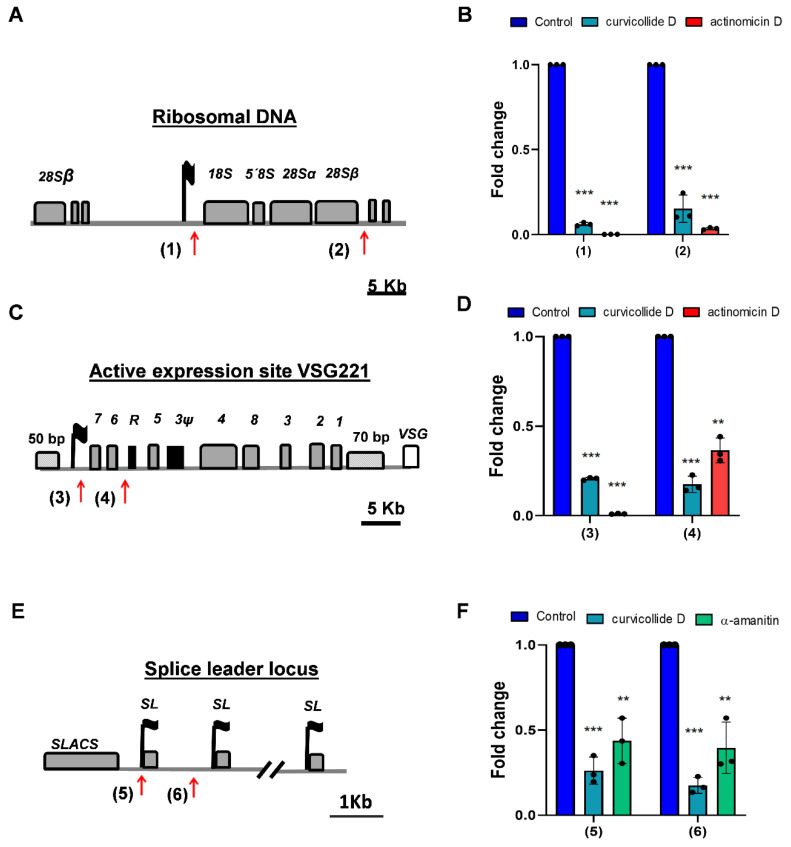
Inhibition of RNA Pol I and Pol II transcription by curvicollide D in bloodstream forms of *T. brucei*. (**A**) Schematic ribosomal DNA (rDNA) transcription unit in *T. brucei*, indicating the location of the primer sets (1) and (2). (**C**) The *VSG221* expression site and (**E**) the splice leader locus with the promoters indicated as a black flag. Genes are shown with grey boxes. *T. brucei* 221 was treated with 1.5 μM of curvicollide D for 3 h. (**B**) Pol I RNA precursor transcripts were analysed in the Pol I transcribed rDNA region (primer pairs (1) and (2)) and so was (**D**) the active *VSG221* ES (primer pairs (3) and (4)). (**F**) RNA pol II precursor transcripts were analysed in the splice leader locus using primer pairs (5) and (6). Actinomicin D and α-amanitin were used as positive controls of inhibition for RNA Pol I and Pol II transcription respectively, and cells treated with DMSO as a negative control. Error bars represent the standard deviation from three independent experiments. Statistical significance was **, *p* < 0.01; ***, *p* < 0.001 using ANOVA and Bonferroni post hoc test.

**Figure 7 ijms-23-06107-f007:**
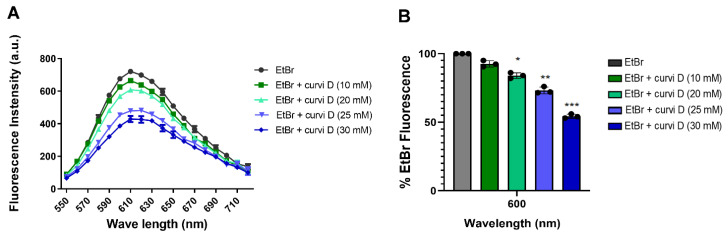
Curvicollide D intercalates into DNA. Ethidium bromide displacement assay using a double stranded DNA plasmid. Increasing concentration of curvicollide D (10 μM to 30 μM) were used to determine (**A**) the fluorescence intensity (arbitrary unit (a.u)) from 550 nm wavelength to 720 nm, and (**B**) the percentage of EtBr fluorescence at 610 nm determined in (**A**). Mean values of triplicates and corresponding standard deviations are shown. Statistical significance between control (EtBr) and each treatment was stimated using Student’s two-tailed *t*-test. *p*-Values were expressed as follows: * *p* < 0.05; ** *p* < 0.01 and *** *p* < 0.001.

**Table 1 ijms-23-06107-t001:** NMR data for curvicollide D in (500 MHz, CDCl_3_, 24 °C).

Curvicollide D		
Position	*δ*_C_, Type	*δ*_H_, mult. (*J* in Hz)
1	29.8, CH_3_	2.22, s
2	213.2, C	
3	52.3, CH	2.69, dq (7.7, 7.2)
3′	14.0, CH_3_	1.10, d (7.2)
4	75.1, CH	4.25, dd (7.7, 7.7)
5	129.0, CH	5.65, dd (15.6, 7.7)
6	136.4, CH	6.32, d (15.6)
7	138.5, C	
7′	13.2, CH_3_	1.83, d (0.9)
8	126.0, CH	5.38, d (9.2)
9	48.4, CH	3.26, dd (11.7, 9.2)
9′	176.2, C	
10	42.7, CH	2.18, m
10′	14.8, CH_3_	1.06, d (6.6)
11	90.4, CH	4.37, d (9.9)
12	131.1, C	
12′	11.6, CH_3_	1.78, d (0.6)
13	129.4, CH	6.11, d (10.8)
14	122.2, CH	6.46, dd (15.3, 10.8)
15	143.0, CH	5.84, d (15.3)
16	73.1, C	
16′	28.0, CH_3_	1.32, s
17	42.6, CH_2_	1.56, m
18	26.2, CH_2_	1.30, m
19	23.1, CH_2_	1.31, m
20	14.0, CH_3_	0.90, t (6.7)

**Table 2 ijms-23-06107-t002:** qPCR primers.

Forward Primer Sequence (5′ → 3′)	Reverse Primer Sequence (5′ → 3′)
(1) rDNAProm_938s: 5′-ATAAAAGGGAGTTATAGCGT-3′	rDNAProm_1048as: 5′-GTACAACACAATCCGTTAAG-3′ [60]
(2) rDNASp 27s _27s: 5′-TATGTGTATGTGTGTTGTGTTA-3′	rDNASp108as_108as: 5′-ATGCAAAATAGGAGACTACA-3′ [60]
(3) ESProm_330s: 5′-GAGATTGTGAGGGTTAGGA-3′	ESProm_662as: 5′-CATCATCCTGCGTCGTTC-3′ [60]
(4) ES_6-F: 5′-GTACAAGCTACGAAAACGTG-3′	ES_6-R: 5′-CCACTCCCACTGGAAACTTA-3′
(5) SL_sp_1245s: 5′-CTTTGTTTCCCATAAGTCTAC-3′	SL_sp_1306As: 5′-AGACACTTGCCATATTTTACT-3′ [60]
(6) SL_spacer_644s: 5′-GCAGCAATAACAGCGAGCATAC-3′	SL_spacer_763As: 5′-GGACGGTTGAGCTGAGTGTAAT-3′ [60]

## Data Availability

Not applicable.

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
