# Peer review of "Curvicollide D Isolated from the Fungus Amesia sp. Kills African Trypanosomes by Inhibiting Transcription"

_ijms, 2022, doi:10.3390/ijms23116107_

Round 1
Reviewer 1 Report
This manuscript describes the characterisation of a natural compound, discovered through HTS of 2000 microbial compounds. The screen yielded a compound, with high anti-trypanosomal activity, that was further explored. Structural characteristics of the compound, determined using a variety of chromatography and mass spectroscopy techniques, showed high similarity to known curvicollides and the novel compound was designated as curvicollide D. The trypanocidal activity of purified curvicollide D was investigated using an Alamar Blue (resazurin) assay and PI flow cytometry. The compound was shown to have an IC50 of 1µM with a selectivity index of 16, both within the DNDi limits for drug selection, and induce cell cycle arrest. The manuscript then considers the mode of action of curvicollide D. The authors show the development of morphological changes and nucleolar disruption in trypanosomes following 3 hours of exposure to the compound and demonstrate a significant reduction in the transcription of RNA Pol I (and VSG-ES) and Pol II. Curvicollide D was then shown to intercalate into dsDNA, a mechanism recognised in several anti-parasitics.
Overall, the manuscript provides solid evidence characterising the novel molecule together with its trypanocidal properties and potential mode of action. The discovery of novel trypanocidal agents is required to ensure the continued improvement of currently available treatments and as such this paper should be published.
There are a few minor issues with the paper that should be addressed prior to publication
- Change the comma to a point in all volumes etc throughout the manuscript (eg line 162; 0,09 to 0.09)
- All gene names should be italicised.
- Remove the underscore from degree (eg line 395; 22oC to 22oC)
- Line 46; Typo meningo-encephalitic
- Line 59; VSG is not the abbreviation for major surface glycoprotein, please reword
- Line 59; delete relying
- Line 102 entered as a query into… There are a few other instances where the grammar is not quite correct, but these will likely be picked up during the editing process.
- Line 162; change to 1 +09 µM
- Line 180; can you add a reference for ‘typical’ distribution for BSF trypanosomes
- Line 183-185; are stats available to confirm the ‘increase’?
- Line 215; consider using an ANOVA / GLM to analyse the data. This would also allow you to see whether the increase in abnormalities was significant between timepoints
- Line 227; ‘slightly lesser’ is a vague term. Was this difference significant?
- Figure 6 could be improved. The labels used can be confused with the panel labels and there are discrepancies between upper and lower case letters in the figure legend.
- Line 347; change Inhibition to inhibition
- Line 411-412; this requires clarification
- Section 4.10.1; Primers. This would be better presented as a table. What cycling parameter were used?
Author Response
Point 1: Change the comma to a point in all volumes etc throughout the manuscript (eg line 162; 0,09 to 0.09).
Done
Point 2: All gene names should be italicised.
Done
Point 3: Remove the underscore from degree (eg line 395; 22oC to 22oC)
Done
Point 4: Line 46; Typo meningo-encephalitic
Done
Point 5: Line 59; VSG is not the abbreviation for major surface glycoprotein, please reword
Now in the manuscript: “variant surface glycoprotein (VSG)”
Point 6: Line 59; delete relying
Done
Point 7: Line 102 entered as a query into… There are a few other instances where the grammar is not quite correct, but these will likely be picked up during the editing process.
Point 8: Line 162; change to 1 +09 µM
Done
Point 9: Line 180; can you add a reference for ‘typical’ distribution for BSF trypanosomes.
We have added a reference where in a previous work we showed this distribution. This reference was already in the previous version of the manuscript. Before reference 59 now 19.
Point 10: Line 183-185; are stats available to confirm the ‘increase’?
Yes, in the new version of the manuscript we have added a new graphic with the statistical significance that confirm the increase (Figure 4d).
Point 11: Line 215; consider using an ANOVA / GLM to analyse the data. This would also allow you to see whether the increase in abnormalities was significant between timepoints
Done. We added the graphic (Figure 5a) using ANOVA and comparing abnormalities between timepoints. We found statistical differences not only between controls and 6 and 15 hours but also between 3hour and 6 hours and 6 and 15 hours of treatment.
Point 12: Line 227; ‘slightly lesser’ is a vague term. Was this difference significant?
We reworded this sentence in the manuscript by “Incubation of trypanosomes with 1.5 μM curvicollide D for 3 hours significantly repressed the expression of the rRNA precursor transcripts, as detected with actinomycin D, a known inhibitor of transcription (Figure 6b).”
Point 13: Figure 6 could be improved. The labels used can be confused with the panel labels and there are discrepancies between upper and lower case letters in the figure legend.
Done. We have changed the letters used to name the pairs of primers by numbers.
Point 14: Line 347; change Inhibition to inhibition
Done
Point 15: Line 411-412; this requires clarification 412-413
Done. In the previous version:
Elution was carried out using isocratic conditions of 5% solvent B for 1.5 min and then a linear gradient from 5% to 100% solvent B in 30.5 min to afford 1 (1.5 mg, tr = 412 25 min) responsible for the observed antiparasitic activity.
In the new version:
Elution was carried out using isocratic conditions of 5% solvent B for 1.5 min and then a linear gradient from 5% to 100% solvent B in 30.5 min to afford curvicollide D (1.5 mg, tR = 25 min), responsible for the observed antiparasitic activity.
Point 16: Section 4.10.1; Primers. This would be better presented as a table. What cycling parameter were used?
Done. We have included a table “Table 2” (line 487) in “materials and methods” section with the primers used in Figure 6, and the cycling parameters for qPCR: 40 cycles of (50 °C, 2 min; 95 °C, 10 min; 95 °C, 15seg; 60 °C, 1 min)
Reviewer 2 Report
The manuscript presented by Ortiz-Gonzalez and collaborators presents an important finding regarding the antiparasitic activity of Curvicollide D against the protozoan Trypanosoma brucei. The manuscript is very well presented and presents important data on antiparasitic activity and possible mechanism of action.
I would like to make a minor suggestion and comment. On page 13 (lines 480-494), I think the authors could better describe the primers in a table and describe, in this session, which genes/targets will be amplified/quantified by qPCR. I think that this information is not clear to readers as it is presented only in the results/discussion of the manuscript.
Author Response
Point 1: I would like to make a minor suggestion and comment. On page 13 (lines 480-494), I think the authors could better describe the primers in a table and describe, in this session, which genes/targets will be amplified/quantified by qPCR. I think that this information is not clear to readers as it is presented only in the results/discussion of the manuscript.
Done. We changed the letters to name the primers by numbers. In addition, a table, “Table 2” with the primers have been included in the “materials and methods” section. Line 487
We added the following explanation:
“Sets of primers (1) and (2) are located in the ribosomal DNA. Primers (3) and (4) are located within the active expression site VSG221. Sets of primers (3) and (4) are located within the splice leader locus. All primers were located in intergenic regions”.